# Dietary Intake of Vegan and Non-Vegan Endurance Runners—Results from the NURMI Study (Step 2)

**DOI:** 10.3390/nu14153151

**Published:** 2022-07-30

**Authors:** Katharina Wirnitzer, Karl-Heinz Wagner, Mohamad Motevalli, Derrick Tanous, Gerold Wirnitzer, Claus Leitzmann, Thomas Rosemann, Beat Knechtle

**Affiliations:** 1Department of Research and Development in Teacher Education, University College of Teacher Education Tyrol, 6010 Innsbruck, Austria; seyed.motevalli-anbarani@student.uibk.ac.at (M.M.); derrick.tanous@student.uibk.ac.at (D.T.); 2Department of Sport Science, Leopold-Franzens University of Innsbruck, 6020 Innsbruck, Austria; 3Research Center Medical Humanities, Leopold-Franzens University of Innsbruck, 6020 Innsbruck, Austria; 4Department of Nutritional Sciences, University of Vienna, 1090 Vienna, Austria; karl-heinz.wagner@univie.ac.at; 5AdventureV & Change2V, 6135 Stans, Austria; gerold@wirnitzer.at; 6Institute of Nutrition, University of Gießen, 35390 Gießen, Germany; claus@leitzmann-giessen.de; 7Institute of Primary Care, University of Zurich, 8091 Zurich, Switzerland; thomas.rosemann@usz.ch (T.R.); beat.knechtle@hispeed.ch (B.K.); 8Medbase St. Gallen Am Vadianplatz, 9001 St. Gallen, Switzerland

**Keywords:** nutrition, dietary assessment, dietary patterns, food frequency, plant-based diet, health, lifestyle, endurance running, marathon, athletes

## Abstract

Nowadays, the growing popularity of distance running has been accompanied by the increasing prevalence of vegan and vegetarian diets, especially among endurance athletes. The present study aimed to examine the association between diet type and dietary intake of distance runners competing at distances longer than 10 km. From a total of 317 participants, 211 endurance runners (57% females) were considered the final sample after applying the exclusion criteria. Runners were assigned to three groups based on the self-reported diet types: 95 omnivores, 40 vegetarians, and 76 vegans. Data collection was conducted using an online survey with questions about sociodemographic information, dietary intake, and dietary-associated motives. A comprehensive food frequency questionnaire with 53 food groups (categorized in 14 basic—plus three umbrella—food clusters) was used to assess dietary intake. Vegan runners had a higher intake of “beans and seeds”, “fruit and vegetables”, and “dairy alternatives”, as well as lower intakes of “oils” than other two groups. Vegetarian runners had a lower intake of “dairy products” and “eggs” than omnivores. A greater intake of “alcohol” and a lower intake of “meat alternatives” was observed in omnivorous runners compared to vegans and vegetarians. Despite the existence of a tendency toward the consumption of health-related food clusters by vegan runners, further investigations are needed to verify the predominance of vegans in health-oriented dietary patterns.

## 1. Introduction

Diet type can significantly impact the nutritional status of endurance athletes, thus influencing their health and performance [1,2]. Two major diet type categories are vegan (defined as a diet containing ingredients from 100% plants only) and vegetarian (defined as a diet devoid of meat and flesh foods), which can be adopted for several motives, including performance, health, ethical issues, and environmental aspects [3,4]. In line with the increasing expansion of veganism and vegetarianism trends [5], the prevalence of vegan and vegetarian diets increased over the past years, especially among endurance athletes [6,7]. This growing prevalence of people following some kind of vegetarian diet has reportedly doubled during the past two years since the commencement of the COVID-19 pandemic [8]. In parallel, the popularity of distance running (including 10 km, half-marathon, marathon, and ultra-marathons) had a similar growth during the past decade, and research shows that, currently, more than 100 million people in Western countries take part in recreational and professional distance runs [9]. Together, it has been estimated that at least 10% of distance runners follow vegan or vegetarian diets, with an increasing popularity of plant-based diets among endurance runners and a higher trend in runners of longer distances such as ultramarathoners [2,10].

A crucial part of sports nutrition practice is to improve, optimize, and personalize exercise-related dietary strategies. With this, dietary assessment helps identify exercise-induced nutritional undersupply that may result in deficiencies (commonly occurring due to higher energy and nutritional needs and/or following restrictive diets). The nutritional requirements of distance runners are more serious than other runners due to the ridged training routine (high mileage and intensity) and racing activities [11]. Evidence indicates that the majority of both recreational [12] and elite [13] endurance athletes are at risk of low bioavailability of energy supply, which mainly originates from their poor dietary habits [14], independently of adherence to some kind of diet. However, an insufficient supply of calories and nutrients may be a result of gastrointestinal distress, the inability to match meal frequencies with training and racing schedules (of travelling athletes), and/or poor nutritional knowledge of dietary choices [15,16].

This concern seems to be more critical for vegan and vegetarian athletes if their diet is not planned appropriately [17]. In this regard, consuming diverse groups of plant foods has been recommended to vegan/vegetarian athletes to meet nutritional guidelines required for optimizing health and performance [1,4,7]. Despite the lower availability of some nutrients (e.g., protein, calcium, zinc, iron, iodine, and vitamin B_12_) in plant vs. animal-based foods [3,18], investigations on general populations indicate that vegetarians, but especially vegans, had higher intakes of carbohydrates, dietary fiber, antioxidants, phytonutrients, and micronutrients, including magnesium, potassium, folic acid, and vitamin C, but lower intakes of total calories, protein, fat (total and saturated fatty acids), and no cholesterol compared to omnivores [4,19,20,21]. The high-fiber and high-nutrient density characterization of, especially, wholesome, vegan diets [22,23] may result in favorable advantages in endurance performance if the diet is planned appropriately [1,2]. However, it has been well-documented that nutrient deficiencies resulting from plant-based diets are largely due to the poor application of nutrition knowledge and meal planning and thus cannot be the overarching diet type per se [22,24,25].

Surveys on endurance runners have also reported that vegans/vegetarians had a higher reliance on carbohydrates and a lower reliance on fat and protein when compared to omnivorous runners [26,27]. Regarding supplement intake, data demonstrate that vegan endurance runners consume more vitamin, but not mineral and macronutrient, supplements compared to both omnivorous and vegetarian runners [27]. In general, it has been shown that vegan and vegetarian runners have higher scores of diet quality compared to omnivorous runners [2]. Moreover, vegan runners appear to have better food choice scores than runners who follow an omnivorous diet [28].

To date, there are limited data on nutritional intakes of athletes based on dietary subgroups [27,28]. This gap in the literature can be partially due to the lack of discrimination between vegan and vegetarian diets in data analysis or an inadequate number of vegan and vegetarian participants in a specific athletic sample. Available research investigating dietary intakes of endurance athletes did not discriminate for different diet types [29,30,31,32] or running distances [33]. Given the importance of these variables in sports nutrition, it seems crucial to scrutinize and compare the dietary patterns of endurance runners to provide novel information for the practical application of individualized nutritional training/racing strategies. Therefore, the objective of the present study was to assess the dietary profiles of vegan, vegetarian, and omnivorous distance runners to identify the potential differences associated with their sociodemographic characteristics. It was hypothesized that runners with a vegan or vegetarian diet tend to consume healthier foods/dietary patterns than omnivores.

## 2. Materials and Methods

### 2.1. Study Design

The present cross-sectional investigation is part of the Nutrition and Running High Mileage (NURMI) Study Step 2. The study protocol [34] was approved by the ethical board of St. Gallen, Switzerland (EKSG 14/145; May 2015) with the trial registration number: ISRCTN73074080. The methods of the “NURMI Study Step 2” have been previously explained in further detail elsewhere [28,35].

### 2.2. Study Participants and Experimental Approach

Endurance runners were mostly from Germany, Austria, and Switzerland, and were recruited through personal contacts via social media, online running communities, websites of organizers for marathon events, email lists of running magazines, as well as other multi-channel sources. In the NURMI Study Step 2, endurance runners were asked to complete an online survey available in English and German (https://www.nurmi-study.com/en; accessed on 8 July 2022). A written description of the procedures was provided for participants, and they agreed to the informed consent before filling in the questionnaires. There were four initial inclusion criteria for participation in the NURMI Study Step 2, including (1) written informed consent, (2) ≥18 years of age, (3) completion of the questionnaire, and (4) successful participation in a distance running event (at least half-marathon) within the past two years.

Endurance runners were classified according to the race distance and their self-reported diet types with a minimum of six months of adherence. Three diet-based groups were defined, including omnivorous (or Western diet, with no food restriction), vegetarian (devoid of meat and all flesh foods, including seafood), and vegan diet (devoid of all types of foods from animal sources, including eggs, dairy products, and honey) [3,4]. The initial race distance subgroups were “half-marathon” and “marathon/ultra-marathon”. Marathoners and ultra-marathoners were pooled in the same group since marathon distance is usually included in an ultra-marathon event. The shortest ultra-marathon distance reported by runners was 50-km and the longest was 160-km. A total number of 74 runners were recognized to complete the 10-km distance, but they had not successfully participated in either a half-marathon or a marathon. Those who met the inclusion criteria (1) to (3) were added as another race distance subgroup since they had provided provided answers comparable to runners who compete in half-marathon, marathon, and ultramarathon events.

### 2.3. Data Clearance

From the initial number of 317 participants, 106 runners were excluded from the sample (including 46 runners who did not meet the inclusion criteria). The Body Mass Index (BMI) approach, following the World Health Organization (WHO) guidelines [36,37], was considered in order to control for a minimal status of health associated with a minimum level of fitness and thus further increase the reliability of data sets. Accordingly, one participant with a BMI greater than 30 kg/m^2^ was excluded from the sample. Further, based on the exclusion criterion of consumption of ≤50% carbohydrates in their total daily energy intake (which is lower than the minimum level recommended for maintaining health-performance association [38,39], we excluded an additional number of 25 runners from the sample. In addition, 34 participants with inconsistent statements on water intake (e.g., never drinking water) were excluded from the study to avoid contradictory data on dietary intake [38]. Finally, in order to control for accuracy in assignment to dietary subgroups, 24 runners (11% of the final sample) had to be shifted to other dietary subgroups (2 vegans to omnivores, 2 vegans to vegetarians, and 20 vegetarians to omnivores), with an 89% representation (*n* = 187) of the recreational runners adequately assessing their diet type. As a final sample, 211 runners (including 95 omnivores, 40 vegetarians, and 76 vegans) with complete data sets were included for data analysis. Figure 1 shows the enrollment and classifications of participants.

### 2.4. Measures and Analytical Modeling

The validated food frequency questionnaire (FFQ) of the “German Health Interview and Examination Survey for Adults (DEGS)” (DEGS-FFQ; with permission of the Robert Koch Institute, Berlin, Germany) [40,41] was used for the present study. Participants were asked to state their regular food intake, particularly in the past four weeks, based on the frequency of consumption (single-choice out of 11 options ranging from “never” to “5 times a day”) and quantity of dietary items (single-choice with various options depending on the food group), including meals eaten in restaurants, canteens, or at parties.

Based on the 53 food groups of the DEGS-FFQ and according to the NOVA classification system developed by the Food and Agriculture Organization (FAO) of the United Nations (UN) [42,43,44,45], food groups were categorized with the corresponding questions pooled for a total of 17 food clusters in order to conduct quantitative and qualitative analysis (Table 1). Self-reported data were linked to diet type groups, including sociodemographic information, general motive(s) for adhering to diet type, and food frequency data.

### 2.5. Statistical Analysis

The statistical software R version 4.1.1 Core Team 2018 (R Foundation for Statistical Computing, Vienna, Austria) was used to conduct all statistical analyses. Exploratory analysis was performed by descriptive statistics, including mean values, standard deviation (SD), median, and interquartile range (IQR). 

Univariate tests were used to investigate the differences between diet type. Chi-square test (χ^2^, nominal scale) was used to examine the association of diet type with sex, nationality, academic qualification, marital status, race distance, and dietary motives. Kruskal–Wallis tests (ordinal and metric scale) were conducted (by using the F distributions) to examine the association of diet type with age, body weight, height, and BMI. 

Food clusters were defined by 53 manifest parameters (assessing how often and level of consumption of specific dietary items). In order to scale dietary intake displayed by measures, items, and clusters, a heuristic index (as a new compound variable) ranging between 0 and 100 was defined (equivalence in all items), and FFQ was calculated by multiplying the two questions and dividing by the maximum. Examination of significant differences in the intake of specific food clusters by diet type and age was conducted by a linear-regression model. The regression analysis assumptions have been confirmed by inspection of graphs of residuals. Differences in respective food clusters between dietary subgroups are also displayed by effect plots with the standardized regression coefficient (β) and 95% confidence interval (CI).

The statistical significance level was set at *p* ≤ 0.05.

## 3. Results

A total number of 211 runners (including 95 omnivores, 40 vegetarians, and 76 vegans) with a median age of 38 years (IQR 18) and a median BMI of 21.7 kg/m^2^ (IQR 3.4) were considered as the final sample for statistical analysis. Germany, Austria, and Switzerland (i.e., D-A-CH countries) had the most participants (96% of the final sample), and the remaining runners were from other countries across the globe. Significant differences were found between dietary subgroups in age (*p* = 0.040), sex (*p* = 0.013), weight (*p* = 0.002), and BMI (*p* = 0.001). There were no significant differences in height, nationality, marital status, academic qualification, and race distance between omnivorous, vegetarian, and vegan runners. Table 2 shows the runners’ sociodemographic profile across dietary subgroups.

Table 3 displays the list of diet-related motives and the associated differences between omnivorous, vegetarian, and vegan runners. Regardless of diet type, “health and wellbeing” (by 85%), “animal ethics” (inclusive animal welfare and animal rights) (by 78%), and “ecological aspects” (inclusive protection of environment and climate) (by 73%) were the most prevalent motives reported by runners as the main reasons to follow their diet types. Regarding the top 3 motives, while the same findings with the general sample were found for vegetarians, different results were observed for omnivorous and vegan runners (Table 3). Significant differences between dietary subgroups were found in five motives, including “sporting performance” (*p* = 0.028), “food scandals” (*p* = 0.010), “animal ethics” (*p* < 0.001), “ecological aspects” (*p* = 0.005), and “custom/tradition” (*p* = 0.007).

Significant differences were found between omnivores, vegetarians, and vegans in the consumption frequency of 11 food clusters (*p* < 0.05). While vegans reported a more frequent intake of “beans and seeds” (*p* < 0.001), “fruit and vegetables” (*p* < 0.001), dairy alternatives” (*p* < 0.001), “meat alternatives” (*p* < 0.001), and “protein” (*p* < 0.001), consumption of “oils” were lower (*p* < 0.001) in vegans compared to omnivores and vegetarians. The consumption frequency of “dairy” and “eggs” were higher in omnivores than vegetarians (*p* < 0.001). Omnivores also reported a greater intake of “alcohol” compared to vegetarians and vegans (*p* = 0.004). No significant difference between diet groups was observed in “grains”, “snacks”, “water”, “beverages”, “processed foods”, or “free/added sugar” (*p* > 0.05). Table 4 displays the differences in food frequency clusters and the subset items among omnivorous, vegetarian, and vegan runners.

Figure 2 shows differences in food clusters between dietary subgroups. Additional details about the regression results, including *p*-values, are displayed in Table 5. Assuming the omnivorous diet as the reference group, the vegetarian diet was a significant predictor of 11 food clusters (*p* < 0.05), while the vegan diet was a significant predictor of 15 food groups (*p* < 0.05). Results indicate that age was a significant predictor of two clusters, including “fruit and vegetables” (*p* = 0.024) and “meat alternatives” (*p* = 0.011).

## 4. Discussion

The present investigation examined dietary intake of distance runners (based on 14 basic and three umbrella food clusters) differentiated by vegan, vegetarian, and omnivorous groups. Runners’ motives to follow diet types and the associations between dietary intake and sociodemographic information were also investigated. The essential results showing significant differences within dietary subgroups were that (1) vegan runners had a higher intake of “beans and seeds”, “fruit and vegetables”, and “dairy alternatives”, as well as a lower intake of “oils” compared to vegetarians and omnivores; (2) vegetarian runners had a lower intake of “dairy products” and “eggs” than omnivores; (3) omnivorous runners had greater intake of “alcohol” and a lower intake of “meat alternatives” compared to vegans and vegetarians. In addition, further important findings show that, independent of diet type, (4) “health and wellbeing”, “animal ethics”, and “ecological aspects” were identified as the most popular motives to follow the self-reported kind of diet; and (5) “diet type” has been found to be a predictor for consumption of the majority of food groups, but “age” has been shown to be a predictor of only two food clusters, including “fruit and vegetables” and “meat alternatives”. The existence of some health-related orientations in consumption of some food clusters by vegan runners seems to partially verify the present hypothesis on the positive trend of more frequent consumption of the food groups characterized by beneficially contributing to a health-promoting dietary intake. To the best of the authors' knowledge, this is the first investigation performed on endurance runner dietary intake–food consumption based on the frequency of specific food items–focusing on the strict examination of and comparison between vegan and non-vegan groups.

As a central method to identify nutritional inadequacy, dietary assessment can help to develop personalized dietary strategies for improving health and performance levels in athletes and optimizing health-related insights in general populations [46,47]. Dietary records, 24-h dietary recalls, the food frequency questionnaire (FFQ), and in-depth interviews are common dietary assessment methods in clinical and scientific approaches [46,47,48]. Data show that dietary recalls, food records, and detailed interviews take a lot of time and energy to be conducted precisely in athletes [49,50]. Compared to these methods, however, FFQs have been found to be a simple, low-cost, and fast method with less burden on target populations [51]. It has been reported that the FFQ is considered the most appropriate survey method to evaluate the dietary intake of athletic populations [51,52]. Athletes, particularly those with unbalanced and/or restrictive diets, are at a higher risk of low energy availability compared to sedentary people [17,53]. In line with the importance of diet regarding health and performance, assessing/monitoring dietary intake is the first and most important step in sports nutrition practice and any personalized sports nutrition counseling [50].

In line with the increasing prevalence of athletes who follow vegan/vegetarian diets, it has been suggested that well-designed plant-based diets can fulfill the nutritional requirements considering health and a successful endurance performance [1,2,54,55,56]. It has been estimated that about 10% of marathoners follow plant-based diets [10]; however, the present study took advantage as it was performed on a greater portion of vegans (36%) and vegetarian (19%) runners, which is statistically reasonable to be compared with omnivores (45%). General results from the present study suggest that the differences between vegans, vegetarians, and omnivores in dietary intake are not limited to foods from animal sources, as differences in plant consumption can potentially exist across the groups, too. These differences may primarily originate from different knowledge, attitudes, and beliefs regarding health and lifestyle between vegan, vegetarian, and omnivorous populations [1,57]. Dietary-related differences between endurance runners with different diet types have also been recognized in patterns of supplement intake, particularly where vegan runners were found to consume more micronutrient supplements than vegetarians and omnivores [27]. 

Results on hydration habits showed that diet type were identified as a non-influencing variable in the consumption of water and beverages. However, there were significant between-group differences in the consumption of milk (predominantly omnivores) and vegetable juices (predominantly vegans) in the present study. While little is known about the hydration status of vegan and vegetarian athletes, it seems that training and racing behaviors play a stronger role in the hydration-related dietary behaviors of endurance athletes [58]. Concerning the prevalence of alcohol intake, significant associations were observed in the present study, where omnivorous runners reported a greater intake of alcohol than vegans and vegetarians. However, this finding might be associated with the unbalanced sex distribution of the study groups, particularly the higher number of males in the omnivorous group and females in the vegetarian and vegan groups. In this regard, it has been documented that the prevalence of alcohol intake is 2- to 3-fold higher in male athletes [59,60] and non-athletes [61] compared to their female peers. Inconsistent with the present results on dietary intake, similar studies considering runner state of health reported that vegan, vegetarian, and omnivorous runners may have similar attitudes regarding the avoidance of alcohol intake [28].

In the present study, no difference in grain consumption was observed between vegan, vegetarian, and omnivorous runners; however, differentiated results showed a significant association between diet type and refined (but not whole) grains, where vegan runners had a lower consumption of refined grains than vegetarians and omnivores. Vegan runners were also found to have a higher intake of two other carbohydrate sources, including “beans and seeds” and “fruit and vegetables”. Available literature shows that vegans and vegetarians generally consume a higher amount of carbohydrates than omnivores [20,21]. For ultra-endurance activities, a greater proportion of daily carbohydrates may warrant the associated needs for performance and recovery [62]. Evidence shows that distance runners supply 60–80% of their energy needs from carbohydrates, while the majority of carbohydrates come from plant-based foods [63,64]. While data regarding the predominance of a specific diet type on endurance performance is limited, evidence has shown that even ultra-endurance sporting performance can be successfully and healthfully completed on a high-carbohydrate vegan diet [65,66].

Interesting findings were observed concerning the protein-based clusters, which are not associated with the characteristics of dietary subgroups. Vegans, followed by vegetarians, reported a higher intake of both “dairy alternatives” and “meat alternatives” compared to omnivores. While these findings may be considered compensating behaviors to reach the required amount of protein intake, evidence indicates that the proportion of protein and fat in the daily energy intake of vegans/vegetarians is generally lower than omnivores [20,21,67]. This evidence is consistent with the present finding, where vegan runners (but not vegetarians) had significantly lower consumption of “oils” than omnivores. In athletic populations, recent trends show a higher intake of protein than recommended [68,69], and about 75% of the general protein supply is derived from animal sources [70]. However, marathoners have been reported to consume higher amounts of plant proteins than other athletes and the general population [63,69]. It has also been reported that the lower amount of fat intake in vegan/vegetarian populations seems to be adequate to match the nutritional needs of endurance runners [7,22]. Consistent with the present results, data indicates that a dietary shift toward a lower intake of animal foods and more plant foods is associated with a lower intake of processed meat and high-fat foods, including fast foods [71], thus improving health and performance [1]. In general, while plant-based diets are considered healthier, they need to be well-balanced, including diverse food items and categories to provide the required protein necessary in order to support good health, performance, and recovery [1,17].

Recent evidence indicates that distance runners have acceptable health-related behaviors regarding food choices since most of them prefer to intake healthy dietary ingredients [72]. The present findings show that vegan runners have a lower consumption of “refined grains” and “oils” along with a greater consumption of “beans and seeds” and “fruit and vegetables”, which may allow for some health-related interpretations in their dietary behaviors. However, caution is advised for any conclusion, particularly given the significant unbalanced distribution of males and females among the study groups. In this regard, recent data show that female runners have a healthier dietary intake than males [73]. In addition, it has been reported that females are generally more focused on diet and health, and males prefer participating in physical activity opportunities to maintain a healthy lifestyle [74]. Consistent with the present study, studies on Western populations show that the likelihood of being vegan or vegetarian is twice as high in females compared to males [75]. Regardless of sex, the health-related dietary patterns found in the vegan runners seem to be associated with a greater health-consciousness of those who follow some kind of vegetarian diet [28]. Independent of the diet type, however, it has been reported that distance runners rely more on self-designed dietary strategies for training and competition [76].

Despite the general fact that vegetarians only focus on specific food items among animal sources (i.e., dairy products and eggs), the present findings show that they consume both “eggs” and “dairy products” considerably less than omnivores. This result may suggest that they are in the transition process of becoming vegan and may potentially follow a 100% plant-based diet in the future. Another finding to support this assumption was that vegetarians reported consuming dairy alternatives 77% higher than omnivores but 28% less than vegans, and meat alternatives 3.8 times (380%) higher than omnivores but only 17% less than vegans. While a possible explanation for these findings can be found in living conditions (particularly living with a partner or family member who follows a plant-based diet), it should be considered that the lack of discrimination between vegan and vegetarian diets in most comparable studies has made it difficult to compare and interpret the present findings with the similar investigations. As a well-established fact, personalized nutritional requirements of athletes, which is connected to their different performance-related challenges, is considered a remarkable factor for explaining most dietary contradictions [77].

Educational background and specific knowledge regarding sports nutrition especially may also be associated with improved health behaviors, including a healthier diet [78]. Concerning academic qualification, however, no significant difference across dietary subgroups was found in the present study on runners. Moreover, in the present study, age was not a significant predictor for consumption of most food groups except for two food clusters: “fruit and vegetables” and “meat alternatives”. Available results from dietary studies show that age is a moderate indicator of dietary patterns in general populations [74,79]. However, caution is advised for further interpretations as omnivores runners in the present study were significantly older than vegans and vegetarians. While this unbalanced age distribution can be partially explained by the higher prevalence of vegan/vegetarian diets in younger than older adults [7,80], it is necessary to consider that most participants in this study were recreational runners. The term “professionalism”, which corresponds to a specific performance level, has been shown to be a key indicator of dietary strategies for training and racing independent of age [77,81]. In this regard, the major motivators for recreational athletes to participate in sports events seem to be health and/or hobby, whereas performance and race-related aspects are considered the main motives of professional athletes for participating in running events/activities [82].

The present investigation has some limitations necessary to mention. The study was set up as a cross-sectional design with self-reported data, thus, under- or over-reporting might have occurred on individual data sets. Misreporting of dietary intake is somewhat prevalent in athletic populations [46]. However, multiple control questions were implemented in the survey to control for contradictory data and minimize the level of bias. In this regard, raw data were also checked for congruency and meaningfulness. The heterogeneity of the sample considering sex and age among dietary subgroups, as well as the larger-than-average proportion of vegetarian/vegan participants, may also be considered as another limitation affecting the health-related findings and interpretations. Furthermore, although the validity of a FFQ is an accepted method to assess dietary intake [50,51], especially for athletes [51,52], this method fails to provide details about the macro- and micro-nutrient status of the athletes (on which the majority of nutritional recommendations are based). Despite the aforementioned limitations, findings from the present study markedly contribute to providing novel data to current scientific knowledge regarding the association between diet type and dietary intake among endurance runners competing in different race distances. The present study provides new directions for future interventional studies on athletes regarding contemporary aspects of nutritional differences between vegan, vegetarian, and omnivorous athletes. However, future research on endurance runners with larger samples and more differentiated groups will help to make comparable data available for a better understanding of the dietary patterns of vegan, vegetarian, and omnivore runners. Together, such investigations provide a firm knowledge helping to design and apply nutritional strategies for prolonged adherence to training and competition in a healthy state.

## 5. Conclusions

Analysis of endurance runner dietary intake showed that there are differences between vegan, vegetarian, and omnivorous runners in their food choices (assessed by FFQ, in the form of amount and frequency of consumption of 17 food clusters). Vegan runners had a lower consumption of “refined grains” and “oils” along with a higher consumption of “beans and seeds”, “fruit and vegetables”, and “dairy alternatives” compared to vegetarians and omnivores. These findings suggest that vegans have a higher tendency toward consumption of frequently-recommended food groups in favor of health (e.g., high-fiber foods). However, there are some considerations (e.g., unbalanced sex and age distribution in the study groups) that may potentially obscure any health-related interpretation regarding this result, but it appears that the health-oriented lifestyle of runners who follow plant-based diets can play a remarkable role in many dietary patterns. Considering the growing popularity of distance running and plant-based diets alike, the present findings may be of help for sports dietitians, nutritionists, and coaches to provide more precise dietary recommendations. These results may also produce new aspects for sports nutrition specialists and researchers regarding the hidden health-related dietary patterns of vegan and vegetarian runners, independent of the food items associated with animal- versus plant-based characteristics. However, more detailed interventions using further analyses of interacting factors are necessary to expand this area of knowledge, particularly to find the association between diet type and health-related dietary choices in endurance runners.

## Figures and Tables

**Figure 1 nutrients-14-03151-f001:**
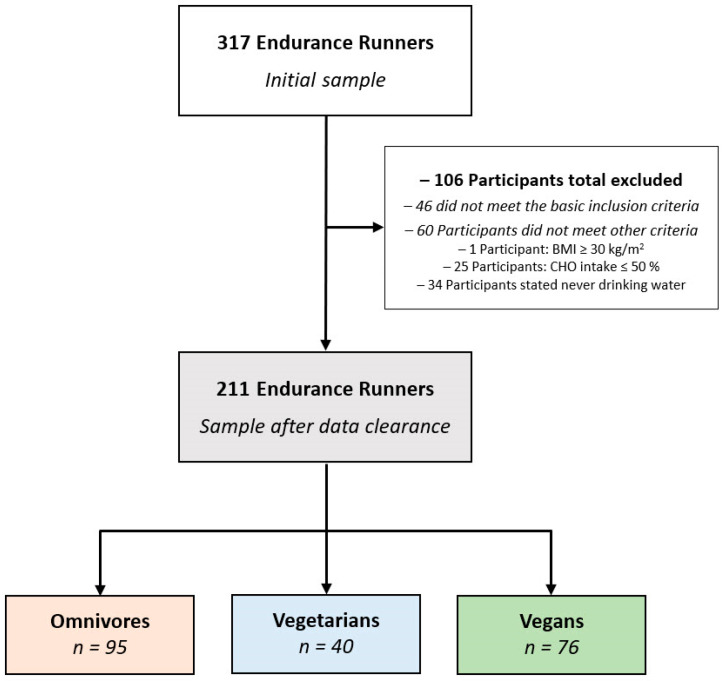
Participants’ enrollment and classifications by kind of diet.

**Figure 2 nutrients-14-03151-f002:**
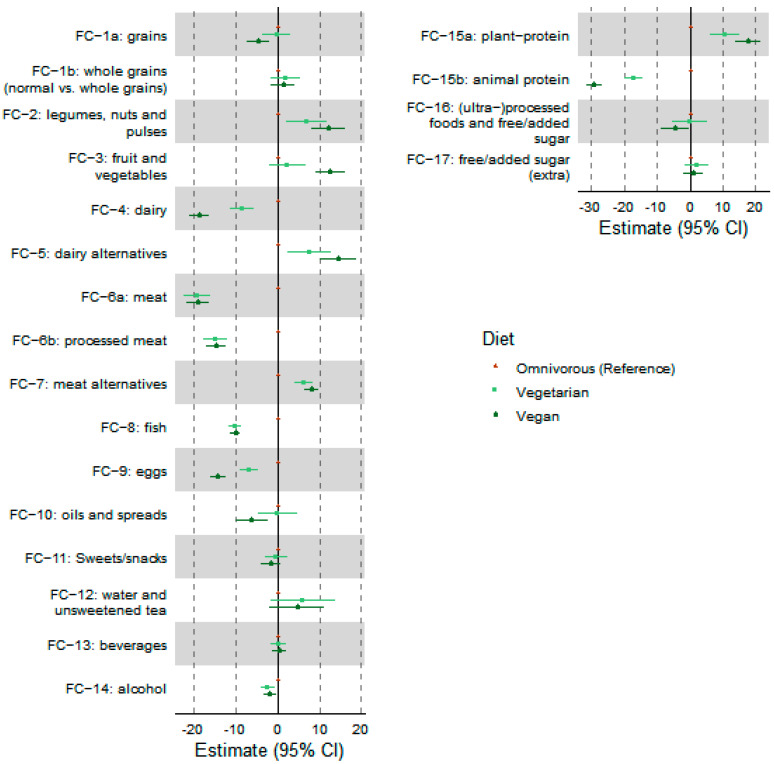
Forest plots with 95% of confidence interval to show the differences between omnivores, vegetarians, and vegans in basic (left column) and umbrella (right column) food clusters. Omnivorous group considered as reference with differences based on variations of vegetarians and vegans from omnivores. FC—food clusters.

**Table 1 nutrients-14-03151-t001:** Modeling of the Food Clusters: 14 Basic Nutrition and Consumption Clusters with 3 Umbrella-Preparation Clusters.

**Basic Food Clusters**		
Cluster 1	Grains	
	a-grains	cornflakes; white bread; white pasta
	b-whole grains	muesli; wholegrain; mixed bread; wholegrain pasta; wholegrain rice; other grains
Cluster 2	Legumes, nuts, and pulses	pulses; nuts and seeds; legumes
Cluster 3	Fruit and vegetables	vegetable juice; fruit; vegetables
Cluster 4	Dairy products	milk; cheese; yoghurt
Cluster 5	Dairy alternatives	milk alternatives
Cluster 6	Meat	
	a-meat	chicken; beef; pork; deer
	b-processed meat	fried nuggets; hamburger; sausage; kebab; pork; processed meat
Cluster 7	Meat alternatives	tofu; seitan; tempeh; etc.
Cluster 8	Fish, shellfish, and seafood	
Cluster 9	Eggs	
Cluster 10	Oils and spreads	butter; margarine; oils
Cluster 11	Sweets and snacks	sweets; snacks; salty snacks
Cluster 12	Water and unsweetened tea	
Cluster 13	Beverages	
Cluster 14	Alcohol	
**Preparation/Umbrella Clusters**		
Cluster 15	Protein	
	a-plant protein	legumes and beans; vegetables; grains (couscous, quinoa); dairy alternatives (e.g., soy products); meat alternatives
	b-animal protein	dairy products; eggs; meat and processed meat products; fish, seafood, and shellfish
Cluster 16	(Ultra-)processed foodsand free/added sugar	sugary carbonated drinks; kcal reduced/artificially sweetened drinks; fruit juice; free sugar in tea; free sugar in coffee; cereals; sweet and savory spreads; margarine; pasta; sweets, cakes, and biscuits; salty snacks, butter; processed meat; processed plant products
Cluster 17	Free/added sugar	Sweet spread; sugary carbonated drinks; fruit juice; free sugar in tea; free sugar in coffee; cereals; sweets, cakes, and biscuits

**Table 2 nutrients-14-03151-t002:** Sociodemographic characteristics of the runners by diet type.

		Totaln = 211	Omnivorousn = 95	Vegetariann = 40	Vegann = 76	Statistics
**Age (years)**		38 (IQR 18)	43 (IQR 18)	38 (IQR 16)	36 (IQR 14)	F_(2, 208)_ = 3.26; *p* = 0.040
**Sex**	Females	57% (121)	46% (44)	65% (26)	67% (51)	χ^2^_(2)_ = 8.64; *p* = 0.013
	Males	43% (90)	54% (51)	35% (14)	33% (25)	
**Body weight (kg)**		65.0 (IQR 14.1)	68.3 (IQR 16.0)	61.0 (IQR 8.1)	64.1 (IQR 11.0)	F_(2, 208)_ = 6.56; *p* = 0.002
**Height (m)**		1.7 (IQR 0.1)	1.7 (IQR 0.1)	1.7 (IQR 0.1)	1.7 (IQR 0.1)	F_(2, 208)_ = 1.80; *p* = 0.167
**BMI (kg/m^2^)**		21.7 (IQR 3.4)	22.5 (IQR 3.5)	20.7 (IQR 3.4)	21.3 (IQR 3.0)	F_(2, 208)_ = 6.83; *p* = 0.001
**Nationality**	Austria	17% (36)	20% (19)	15% (6)	14% (11)	χ^2^_(6)_ = 8.22; *p* = 0.222
	Germany	74% (156)	72% (68)	80% (32)	74% (56)	
	Switzerland	5% (11)	7% (7)	2% (1)	4% (3)	
	Other Countries	4% (8)	1% (1)	2% (1)	8% (6)	
**Marital Status**	Divorced/Separated	5% (11)	3% (3)	2% (1)	9% (7)	χ^2^_(4)_ = 8.45; *p* = 0.077
	Married/Partner	68% (143)	75% (71)	57% (23)	64% (49)	
	Single	27% (57)	22% (21)	40% (16)	26% (20)	
**Academic** **Qualification**	Upper Secondary	33% (69)	37% (35)	40% (16)	24% (18)	χ^2^_(6)_ = 7.88; *p* = 0.445
	A Level or Equivalent	23% (49)	24% (23)	18% (7)	25% (19)	
	University/College	34% (72)	33% (31)	32% (13)	37% (28)	
	No Answer	9% (21)	6% (6)	10% (4)	13% (10)	
**Race Distance**	10 km	35% (74)	36% (34)	30% (12)	37% (28)	χ^2^_(4)_ = 1.41; *p* = 0.843
	HM	39% (83)	38% (36)	48% (19)	37% (28)	
	M/UM	26% (54)	26% (25)	22% (9)	26% (20)	

Data are presented as median and interquartile range (IQR) or prevalence (%; *n*). BMI—body mass index. Km—kilometers. HM—half-marathon. M/UM—marathon/ultra-marathon. Statistical methods: Chi-square tests (χ^2^) and Kruskal–Wallis tests (*F*-values).

**Table 3 nutrients-14-03151-t003:** Motives of runners for adhering to their self-reported diet types and differences between dietary subgroups.

	Total *n* = 211	Omnivorous *n* = 95	Vegetarian *n* = 40	Vegan *n* = 76	Statistics
**Health and Wellbeing**	85% (106)	78% (18)	86% (24)	88% (64)	χ^2^_(2)_ = 1.25; *p* = 0.535
**Sporting Performance**	51% (63)	57% (13)	29% (8)	58% (42)	χ^2^_(2)_ = 7.16; *p* = 0.028
**Food Scandals**	35% (44)	17% (4)	57% (16)	33% (24)	χ^2^_(2)_ = 9.24; *p* = 0.010
**Animal Ethics**	78% (97)	43% (10)	75% (21)	90% (66)	χ^2^_(2)_ = 22.84; *p* < 0.001
**Ecological Aspects**	73% (91)	48% (11)	71% (20)	82% (60)	χ^2^_(2)_ = 10.65; *p* = 0.005
**Social Aspects (world hunger)**	49% (61)	35% (8)	50% (14)	53% (39)	χ^2^_(2)_ = 2.44; *p* = 0.295
**Economic Aspects**	18% (22)	9% (2)	11% (3)	23% (17)	χ^2^_(2)_ = 3.78; *p* = 0.151
**Religion/Spirituality**	6% (8)	-	14% (4)	5% (4)	χ^2^_(2)_ = 4.55; *p* = 0.103
**Custom/Tradition**	5% (6)	17% (4)	-	3% (2)	χ^2^_(2)_ = 9.99; *p* = 0.007
**Taste and Enjoyment**	44% (54)	43% (10)	39% (11)	45% (33)	χ^2^_(2)_ = 0.29; *p* = 0.866
**No Specific Reason**	<1% (1)	4% (1)	-	-	χ^2^_(2)_ = 4.43; *p* = 0.109

**Table 4 nutrients-14-03151-t004:** Differences between vegan and non-vegan runners in food frequency clusters and items.

	Omnivorous *n* = 95	Vegetarian *n* = 40	Vegan *n* = 76	Statistics
**Part A—Basic Clusters**				
** *FC* ** ** *—* ** ** *1: * ** ** *Total of grains* **	18.72 ± 9.18	19.63 ± 7.29	16.77 ± 8.38	F_(2, 208)_ = 2.00; *p* = 0.138
**FC** **—** **1** **a (Total of refined grains)**	14.09 ± 9.68	13.88 ± 8.23	9.39 ± 8.34	F_(2, 208)_ = 7.21; *p* = 0.001
**Cornflakes**	2.04 ± 5.21	0.60 ± 2.27	1.39 ± 3.53	F_(2, 208)_ = 2.53; *p* = 0.082
**White bread**	8.61 ± 8.59	10.05 ± 7.95	5.87 ± 6.66	F_(2, 208)_ = 5.07; *p* = 0.007
**White pasta**	12.59 ± 8.98	12.25 ± 9.87	8.24 ± 8.61	F_(2, 208)_ = 5.97; *p* = 0.003
**FC** **—** **1** **b (Total of whole grains)**	18.81 ± 9.59	20.62 ± 8.13	20.07 ± 9.32	F_(2, 208)_ = 1.11; *p* = 0.332
**Muesli**	15.45 ± 12.68	16.26 ± 13.26	18.05 ± 13.74	F_(2, 206)_ = 0.69; *p* = 0.504
**Whole grain bread**	16.79 ± 9.21	19.60 ± 10.21	14.20 ± 8.05	F_(2, 208)_ = 4.48; *p* = 0.012
**Whole grain pasta**	8.44 ± 8.32	8.90 ± 7.96	12.97 ± 8.89	F_(2, 208)_ = 6.72; *p* = 0.001
**Whole grain rice**	6.36 ± 7.48	6.60 ± 7.32	8.53 ± 7.47	F_(2, 208)_ = 2.36; *p* = 0.097
**Other whole grains**	8.61 ± 8.59	10.05 ± 7.95	5.87 ± 6.66	F_(2, 208)_ = 5.07; *p* = 0.007
** *FC* ** ** *—* ** ** *2: * ** ** *Total of beans and seeds* **	20.76 ± 12.34	27.55 ± 13.78	32.94 ± 13.21	F_(2, 208)_ = 20.72; *p* < 0.001
**Nuts and seeds**	16.72 ± 13.02	19.85 ± 13.32	23.16 ± 12.94	F_(2, 208)_ = 5.55; *p* = 0.004
**Legumes**	11.16 ± 8.55	17.15 ± 9.62	21.08 ± 11.06	F_(2, 208)_ = 29.26; *p* < 0.001
** *FC* ** ** *—* ** ** *3: Total of fruit and vegetables* **	25.71 ± 11.91	28.46 ± 10.32	38.94 ± 11.66	F_(2, 208)_ = 30.9; *p* < 0.001
**Vegetable juice**	4.59 ± 9.87	2.77 ± 6.04	8.28 ± 12.53	F_(2, 208)_ = 6.25; *p* = 0.002
**Fruit**	17.09 ± 8.03	20.30 ± 10.81	21.18 ± 8.88	F_(2, 208)_ = 5.67; *p* = 0.004
**Vegetables**	26.45 ± 11.79	30.23 ± 9.69	38.38 ± 10.93	F_(2, 208)_ = 27.54; *p* < 0.001
** *FC* ** ** *—* ** ** *4: * ** ** *Total of dairy* **	18.33 ± 9.34	10.06 ± 9.58	0.00 ± 0.00	F_(2, 208)_ = 205.89; *p* < 0.001
**Milk**	16.08 ± 11.77	6.45 ± 9.63	0.00 ± 0.00	F_(2, 208)_ = 118.17; *p* < 0.001
**Cheese**	13.38 ± 7.96)	7.97 ± 6.76	0.00 ± 0.00	F_(2, 208)_ = 163.04; *p* < 0.001
**Yoghurt**	13.00 ± 10.31	8.88 ± 10.96	0.00 ± 0.00	F_(2, 208)_ = 83.21; *p* < 0.001
** *FC* ** ** *—* ** ** *5: * ** ** *Dairy alternatives* **	9.68 ± 12.82	17.10 ± 14.77	23.89 ± 15.06	F_(2, 208)_ = 26.05; *p* < 0.001
** *FC* ** ** *—* ** ** *6: * ** ** *Total of meat* **	18.10 ± 12.23	0.00 ± 0.00	0.00 ± 0.00	F_(2, 208)_ = 221.01; *p* < 0.001
**FC** **—** **6a (Total of unprocessed meat)**	19.27 ± 13.02	0.00 ± 0.00	0.00 ± 0.00	F_(2, 208)_ = 188.11; *p* < 0.001
**Chicken**	7.79 ± 6.48	0.00 ± 0.00	0.00 ± 0.00	F_(2, 208)_ = 112.89; *p* < 0.001
**Beef and pork and deer**	16.18 ± 12.12	0.00 ± 0.00	0.00 ± 0.00	F_(2, 208)_ = 163.15; *p* < 0.001
**FC** **—** **6b (Total of processed meat)**	14.98 ± 11.89	0.00 ± 0.00	0.00 ± 0.00	F_(2, 208)_ = 178.73; *p* < 0.001
**Fried nuggets**	4.17 ± 4.10	0.00 ± 0.00	0.00 ± 0.00	F_(2, 208)_ = 113.44; *p* < 0.001
**Hamburger**	2.13 ± 3.18	0.00 ± 0.00	0.00 ± 0.00	F_(2, 208)_ = 31.51; *p* < 0.001
**Sausage**	1.71 ± 3.22	0.00 ± 0.00	0.00 ± 0.00	F_(2, 208)_ = 20.30; *p* < 0.001
**Kebab**	1.92 ± 2.73	0.00 ± 0.00	0.00 ± 0.00	F_(2, 208)_ = 48.12; *p* < 0.001
**Other processed meat**	14.42 ± 13.36	0.00 ± 0.00	0.00 ± 0.00	F_(2, 208)_ = 112.8; *p* < 0.001
** *FC* ** ** *—* ** ** *7: * ** ** *Meat alternatives* **	2.13 ± 3.91	8.17 ± 7.84	9.86 ± 6.02	F_(2, 208)_ = 62.14; *p* < 0.001
** *FC* ** ** *—* ** ** *8: * ** ** *Fish* **	10.12 ± 5.61	0.00 ± 0.00	0.00 ± 0.00	F_(2, 208)_ = 411.81; *p* < 0.001
** *FC* ** ** *—* ** ** *9: * ** ** *Eggs* **	14.32 ± 7.60	7.50 ± 7.40	0.00 ± 0.00	F_(2, 208)_ = 194.67; *p* < 0.001
** *FC* ** ** *—* ** ** *10: * ** ** *Total of oils* **	14.88 ± 13.56	14.73 ± 14.72	8.29 ± 9.82	F_(2, 208)_ = 5.55; *p* = 0.004
**Butter**	10.27 ± 12.83	7.20 ± 12.53	0.00 ± 0.00	F_(2, 208)_ = 42.45; *p* < 0.001
**Margarine**	4.80 ± 8.66	7.60 ± 12.31	8.29 ± 9.82	F_(2, 208)_ = 6.79; *p* = 0.001
**Other oils**	7.25 ± 6.86	7.30 ± 7.42	4.14 ± 4.91	F_(2, 208)_ = 4.56; *p* = 0.012
** *FC* ** ** *—* ** ** *11: * ** ** *Total of snacks* **	11.31 ± 7.40	10.97 ± 7.22	9.73 ± 6.76	F_(2, 208)_ = 0.88; *p* = 0.418
**Sweet snacks**	10.99 ± 6.25	10.58 ± 6.72	8.70 ± 6.74	F_(2, 208)_ = 3.56; *p* = 0.030
**Salty snacks**	6.39 ± 7.50	6.15 ± 8.29	6.13 ± 6.19	F_(2, 206)_ = 0.21; *p* = 0.808
** *FC* ** ** *—* ** ** *12: * ** ** *Total of water* **	32.16 ± 19.75	38.59 ± 23.70	37.46 ± 20.77	F_(2, 208)_ = 1.86; *p* = 0.159
**Water**	57.83 ± 27.05	63.20 ± 28.05	59.53 ± 28.16	F_(2, 208)_ = 0.55; *p* = 0.576
**Unsweetened tea**	18.34 ± 14.91	23.00 ± 20.99	24.91 ± 16.09	F_(2, 208)_ = 4.67; *p* = 0.010
** *FC* ** ** *—* ** ** *13: * ** ** *Beverages* **	13.72 ± 4.83	13.92 ± 4.77	13.99 ± 5.28	F_(2, 208)_ = 0.04; *p* = 0.961
** *FC* ** ** *—* ** ** *14: * ** ** *Alcohol* **	4.87 ± 5.46	2.50 ± 3.66	2.97 ± 4.09	F_(2, 208)_ = 5.78; *p* = 0.004
**Part B—Umbrella Clusters**				
** *FC* ** ** *—* ** ** *15: Total of protein* **	39.78 ± 14.25	31.39 ± 14.12	42.56 ± 12.33	F_(2, 208)_ = 8.12; *p* < 0.001
**FC** **—** **15a—Plant protein**	24.50 ± 12.47	35.26 ± 11.56	42.56 ± 12.33	F_(2, 208)_ = 48.77; *p* < 0.001
**FC** **—** **15b—Animal protein**	29.04 ± 9.34	11.89 ± 9.35	0.00 ± 0.00	F_(2, 208)_ = 422.88; *p* < 0.001
** *FC* ** ** *—* ** ** *16: Processed* ** ** *foods and free/added sugar* **	27.89 ± 14.08	27.82 ± 17.57	23.37 ± 12.48	F_(2, 208)_ = 1.80; *p* = 0.168
** *FC* ** ** *—* ** ** *17: * ** ** *Free/added sugar* **	14.01 ± 9.24	16.16 ± 12.53	14.83 ± 9.04	F_(2, 208)_ = 0.41; *p* = 0.663

Data are presented as Mean ± Standard Deviation. FC—food clusters. Statistical methods: Kruskal–Wallis tests (*F*-values) and Chi-square tests (χ^2^).

**Table 5 nutrients-14-03151-t005:** Linear regression results for age- and dietary-based main effects of food clusters with omnivorous reference group.

	Age			Omnivorous vs. Vegetarian			Omnivorous vs. Vegan		
	β	95%-CI	*p*	β	95%-CI	*p*	β	95%-CI	*p*
**Part A—Basic Clusters**									
**FC—1a: Total of refined grains**	0.05	[1.21, −1.12]	0.939	−0.20	[3.15, −3.54]	0.908	−4.67	[−1.91, −7.44]	0.001
**FC—1b: Total of whole grains**	0.11	[1.31, −1.09]	0.407	1.83	[5.28, −1.62]	0.297	1.31	[4.16, −1.55]	0.368
**FC—2: Total of beans and seeds**	0.16	[1.84, −1.53]	0.855	6.83	[11.66, 2.00]	0.006	12.25	[16.25, 8.25]	<0.001
**FC—3: Total of fruit and vegetables**	−1.71	[−0.23, −3.20]	0.024	2.40	[6.65, −1.86]	0.268	12.52	[16.04, 8.99]	<0.001
**FC—4: Total of dairy**	−0.89	[0.08, −1.87]	0.072	−8.46	[−5.67, −11.25]	<0.001	−18.71	[−16.40, −21.02]	<0.001
**FC—5: Dairy alternatives**	0.69	[2.52, −1.13]	0.455	7.56	[12.80, 2.32]	0.005	14.50	[18.83, 10.17]	<0.001
**FC—6a: Total of unprocessed meat**	0.41	[1.55, −0.73]	0.475	−19.18	[−15.92, −22.45]	<0.001	−19.09	[−16.39, −21.80]	<0.001
**FC—6b: Total of processed meat**	0.47	[1.51, −0.56]	0.369	−14.88	[−11.90, −17.86]	<0.001	−14.78	[−12.31, −17.24]	<0.001
**FC—7: Meat alternatives**	0.94	[1.66, 0.22]	0.011	6.23	[8.29, 4.17]	<0.001	8.12	[9.83, 6.41]	<0.001
**FC—8: Fish**	−0.16	[0.33, −0.65]	0.512	−10.15	[−8.74, −11.56]	<0.001	−10.18	[−9.02, −11.35]	<0.001
**FC—9: Eggs**	0.17	[0.96, −0.61]	0.663	−6.78	[−4.53, −9.03]	<0.001	−14.24	[−12.38, −16.11]	<0.001
**FC—10: Total of oils**	0.81	[2.45, −0.82]	0.328	0.02	[4.71, −4.67]	0.994	−6.25	[−2.37, −10.13]	0.002
**FC—11: Total of snacks**	−0.07	[0.86, −1.00]	0.885	−0.35	[2.32, −3.02]	0.797	−1.61	[0.60, −3.81]	0.153
**FC—12: Total of water**	−1.62	[1.09, −4.34]	0.240	6.09	[13.87, −1.69]	0.124	4.62	[11.06, −1.82]	0.159
**FC—13: Beverages**	0.25	[0.90, −0.40]	0.446	0.25	[2.11, −1.61]	0.789	0.38	[1.92, −1.16]	0.625
**FC—14: Alcohol**	0.16	[0.77, −0.45]	0.603	−2.34	[−0.59, −4.09]	0.009	−1.83	[−0.38, −3.28]	0.014
**Part B—Umbrella Clusters**									
**FC—15a: Plant protein**	−0.86	[0.74, −2.45]	0.291	10.58	[15.14, 6.02]	<0.001	17.70	[21.47, 13.92]	<0.001
**FC—15b: Animal protein**	−0.27	[0.70, −1.25]	0.581	−17.20	[−14.41, −19.99]	<0.001	−29.15	[−26.84, −31.46]	<0.001
**FC—16: Processed foods and free/added sugar**	0.10	[1.96, −1.76]	0.918	−0.05	[5.28, −5.38]	0.987	−4.47	[−0.06, −8.88]	0.047
**FC—17: Free/added sugar**	0.14	[1.43, −1.14]	0.828	2.17	[5.86, −1.51]	0.246	0.87	[3.92, −2.18]	0.573

The omnivorous group is considered the reference. β—regression coefficient. CI—confidence interval. *p*—*p*-value. FC—food clusters.

## Data Availability

The datasets generated and/or analyzed during different stages of the current study are not publicly available, but may be made available upon reasonable request. Participants will receive a brief summary of the results of the “NURMI Study” upon request.

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
