# Peer review of "Dietary Intake of Vegan and Non-Vegan Endurance Runners—Results from the NURMI Study (Step 2)"

_nutrients, 2022, doi:10.3390/nu14153151_

Round 1

Reviewer 1 Report

General comments:

This paper is largely well-written and provides some insight into the dietary strategies used by vegan/vegetarian/omnivorous distance runners. While the work is novel, it is conducted in a smaller sample size that does not appear to be representative of the larger running population.

While I have provided some specific comments below, I do have some key concerns:

-       1. Methodologically, there have been many analyses conducted. I believe a post-hoc analysis is required to reduce the likely risk of type I error.

-       2. When reading this study, it seems (to me, at least) that the author has a heavy bias towards vegan diets being “superior” to other diet types. I believe the language needs to be toned down in sections to provide a more balanced representation of the literature and the results.

3  3. There are obvious instances where the author cited their own work to support statements that are not directly related to their previous findings. This must be rectified.

Specific comments:

Abstract:

Line 23: Please expand/replace “multi-level data clearance.” State why the 106 were removed.

Introduction:

Line 39: not sure what you mean with this opening sentence. It is unclear how diet type is “an important indicator of nutritional requirements in endurance athletes.” Maybe something like “considering the nutritional requirements of distance runners, diet type can have significant impacts on performance”?

Line 42: change “are adopted” to “can be adopted”

Line 78: it is unclear to me how a “low calorie” diet could be beneficial for endurance performance? Please expand upon this point, or revise.

Line 88: Please revise this sentence I think you are overstating the findings of the previous study a little bit. I would suggest saying that runners adhering to a vegan diet appear to have lower body weight and better food choice scores than those following an omnivorous diet. Or you could remove this sentence without impacting the message at all.

Line 95: this is an extremely long sentence. I would suggest revising and making more concise.

Methods:

General methodological comments: a large number of analyses have been conducted. I would suggest running a post-hoc analysis (Holm–Bonferroni, etc.) to reduce the risk of type I error.

Line 142: it is unclear to me why you excluded participants who consumed <50% of their diets coming from carbohydrates – particularly considering that low-carb-high-fat dietary patterns have also become more common in endurance athletes. I think this would be an important group to include in your data set. I also believe this would align with the aim of “to assess the dietary profiles of vegan, vegetarian, and omnivorous distance runners”

Line 159: remove the word “friendly”

Results:

As per comment in methods, some of this may need to be revised after post-hoc correction has been applied.

Discussion:

Line 289: This comment/sentence is not supported by your results. You did not assess endurance performance, so this cannot be stated. Similarly, the citations used to support this claim are narrative reviews that broadly report on mechanistic benefits of vegan/vegetarian diets. To my knowledge, crossover trials comparing vegan vs. omnivorous diets have shown no differences in performance adaptations across a training program. Please revise.

Line 292: what do you mean “took advantage”? Maybe had a higher proportion of vegetarian and vegan runners?

Line 308: I think this line is misleading. The difference is 4.5 vs 2.5 vs 2.9 (which is not double both groups). Can you please contextualise this difference in terms of actual alcohol intake (i.e., 1 more alcoholic beverage per week), as I suspect the intake is quite low across all three groups.

Line 325: I think you could remove this sentence.

Line 330: This addition feels like taking a chance at self-citation to me. I don’t think a single case report strengthens the previous point, which is based off some much larger observational cohorts. Please remove this sentence.

Line 354: two points associated with this sentence. Firstly, I don’t think the way it is written makes sense. Secondly, the claim is not supported by the citation. You state that “Recent evidence indicates that distance runners have acceptable health-related behaviours regarding food choices to get the desired ingredients” – and the cite a paper that suggests half-marathon runners are healthier than other runners? This again feels like a sneaky chance to self-cite. Please remove this setnece.

Line 355: I am not sure the term “healthier” here is appropriate. While refined grains do have associations with disease in various populations, this association mostly disappears when accounting for bodyweight (i.e., health status is dictated by excess adiposity, not be the refined grains themselves). Similarly, replacing saturated fats with various oils has been shown to lower CVD risk – which could be considered “healthier.” Please revise this sentence to better describe your results.

Line 373: please change to “may suggest”

 Line 412: one limitation that needs to be mentioned is related to sample size. While ~270 runners is nothing to sneeze at, the larger than average number of participants following vegetarian/vegan diets means that this data may not be representative of the larger running population – please state this.

Line 416: how/what future intervention studies do you suggest being performed based upon your findings?

Conclusion:

Line 427: please revise this sentence and remove the words “healthier” instead state the main results/ areas of difference.

Author Response

Dear Reviewer #1,

Thank you for your consideration, positive feedback, and the opportunity to revise the manuscript nutrients-1835516 entitled “Dietary Intake of Omnivorous, Vegetarian, and Vegan Endurance Runners – Results from the NURMI Study (Step 2)”, which has led to a significant improvement of the manuscript.

In response to your valuable comments, we provided some evidence along with detailed explanations that directly address each comment. We hope our responses below and revisions throughout the manuscript will satisfy the existing concerns of the reviewer and editors. Changes in the manuscript have been highlighted via “Track Changes”.

Kind Regards,

Katharina Wirnitzer on behalf of the team of authors

Review Report 1

-----------------------------------------------------------------------------------------------------------------------------

Comments mentioned by Reviewer #1 (in blue) and our responses to each comment (in red)

-----------------------------------------------------------------------------------------------------------------------------

General comments

This paper is largely well-written and provides some insight into the dietary strategies used by vegan/vegetarian/omnivorous distance runners. While the work is novel, it is conducted in a smaller sample size that does not appear to be representative of the larger running population.

Answer: We appreciate your kind feedback and consideration. It can be shortly mentioned that we agree a larger sample size could enhance the representativeness of the results and associated interpretations. Compared to similar studies the present sample size cannot be considered small; however, we did not make a statement of the representativity of our sample or findings.

In addition, as is displayed in figure 1, we excluded the data of 106 participants from our final analysis in the process of data clearance in order to reach more valid findings. Another justification can be found in the study sampling, where we tried to have a greater focus on vegan and vegetarian running societies (in our communications to invite participants) in order to have 3 semi-equal study groups. As we all know, the prevalence of vegan/vegetarian diet is normally/generally about 5-10% in adults (or a little higher among endurance runners, as mentioned in the Intro), but in the present study we could be able to recruit a considerable number of vegan (n = 76) and vegetarian runners (n = 40) which seems to be comparable to the omnivorous group (n = 95) from statistical viewpoint.

While I have provided some specific comments below, I do have some key concerns:

  1. Methodologically, there have been many analyses conducted. I believe a post-hoc analysis is required to reduce the likely risk of type I error.

Answer: Thank you for your attentive comment. As the NURMI study is an explorative study (generating hypothesis) and not an experimental/interventional study, a post hoc test was not performed (as not required). The further advanced analysis was performed by a regression analysis (forest plot) by testing multiple contrasts with the omnivorous diet as reference (vegetarian and vegan compared to mixed diet). The assessment was made using the parameters of the regression analysis.

  1. When reading this study, it seems (to me, at least) that the author has a heavy bias towards vegan diets being “superior” to other diet types. I believe the language needs to be toned down in sections to provide a more balanced representation of the literature and the results.

Answer: We respect this concern, but to the best of our knowledge and after rechecking, there was no statement in the script upon the superiority of a vegan diet as the present findings are not able to support such a conclusion.

Moreover, Reviewer 2 generally commented positive on the present paper.

As is mentioned in different parts of the paper using different wordings, any type of diet can fail to supply nutritional requirements of endurance runners if not planned appropriately. Having a look on the manuscript’s contents, it seems that the reviewer’s concern might originate from the statements indicating health-related predominance of vegan diet. In this regard, it should be noted that these statements in the Introduction are based on recently-published studies, and in Discussion is naturally based on our findings.

However, we agree with the respected reviewer that some wording should be expressed with more caution. In this context, it would have been helpful if the respected reviewer would have provided more detailed information (as was provided for other concerns, too) and point out of where the reviewer feels that our wording and tone is going to be heavily biased in order to better identify these phrases and to better address and solve the reviewers concern.

Thus, we edited and reworded the relevant phrases more detailed, differentiated and cautious in order to meet the expert reviewer’s concern.

Furthermore, it seems necessary to mention, that–based on sound theory of more than 10,000 papers of plant-based diets in favor of health (the latter as the prerequisite for exercise training and sports performance) from medicine and nutritional sciences, which is robust scientific evidence and state of the art (only to name the current position statement of the AND, the largest specialized nutrition organization worldwide: Melina V, Craig W, Levin S, Dietetics AoNa (2016) Position of the Academy of Nutrition and Dietetics: Vegetarian Diets. J Acad Nutr Diet 115: 1970-1980)–we used this foundation to introduce the background of issue and discussed our results right related to this body of evidence since the present study for the first time identified differences in the dietary intake between endurance runners following specific diet types. Thank you.

Action taken: we revised several statements in the sections Introduction and Discussion, accordingly.

  1. There are obvious instances where the author cited their own work to support statements that are not directly related to their previous findings. This must be rectified.

Answer: Thank you for bringing out this note to our attention. While we agree with this comment, it should be mentioned that there are limited literature investigating on vegan and vegetarian distance runners, and the majority of them originates from our lab and this team of authors. This can be the main reason for the increased number of self-citations.

Action taken: while we included more relevant references where applicable, several less-relevant references (including 3 self-cited references) were removed from the reference list.

Specific comments

  1. Abstract: Line 23: Please expand/replace “multi-level data clearance.” State why the 106 were removed.

Answer: Thank you for your comment. We agree that “multi-level data clearance” is a little confusing. However, explaining the procedure of data clearance may take one or two extra lines and may not be reasonable (as the abstract is currently a little longer than the suggested word-limit i.e., 200 words).

Action taken: the statement was paraphrased, accordingly.

  1. Introduction:

5.a. Line 39: not sure what you mean with this opening sentence. It is unclear how diet type is “an important indicator of nutritional requirements in endurance athletes.” Maybe something like “considering the nutritional requirements of distance runners, diet type can have significant impacts on performance”?

Answer: Thank you for your comment.

Action taken: We revised the statement using clearer wording.

5.b. Line 42: change “are adopted” to “can be adopted”

Answer: Your attention is appreciated.

Action taken: We revised the statement, accordingly.

5.c. Line 78: it is unclear to me how a “low calorie” diet could be beneficial for endurance performance? Please expand upon this point, or revise.

Answer: Thank you. We agree with this concern. Having a look on the literature referenced, it was found that the phrase “low calorie” belongs only to the previous statement and should be removed. However, it can be generally mentioned that in some cases and situations, low-calorie foods may result in better performance-related outcomes compared to high-calorie foods, and this has been probably the reason that we did not recognize any problem while reviewing.

Action taken: We revised the statement, accordingly.

5.d. Line 88: Please revise this sentence I think you are overstating the findings of the previous study a little bit. I would suggest saying that runners adhering to a vegan diet appear to have lower body weight and better food choice scores than those following an omnivorous diet. Or you could remove this sentence without impacting the message at all.

Answer: Thank you for your attentive comment and suggestion. Since the paragraph is comparing dietary behaviors of vegan/vegetarian runners with omnivores, we agree that it is better to talk about dietary-related findings (e.g., food choice scores) rather health-related reports.  

Action taken: we revised the statement, accordingly.

5.e. Line 95: this is an extremely long sentence. I would suggest revising and making more concise.

Answer: Thank you for your comment.

Action taken: We removed a repetitive part (already presented in the first paragraph) and shortened the statement.

  1. Methods:

6.a. General methodological comments: a large number of analyses have been conducted. I would suggest running a post-hoc analysis (Holm–Bonferroni, etc.) to reduce the risk of type I error.

Answer: Thank you again for your attention. We kindly refer the respected reviewer to our answer to comment 1: as the NURMI study is an explorative study and not an experiment or intervention, a post-hoc analysis is not required and thus was not performed. Moreover/all the more, no ANOVA and therefore also no post-hoc test (Bonferroni correction for the p-value adaptation) was necessary to conduct.

6.b. Line 142: it is unclear to me why you excluded participants who consumed <50% of their diets coming from carbohydrates – particularly considering that low-carb-high-fat dietary patterns have also become more common in endurance athletes. I think this would be an important group to include in your data set. I also believe this would align with the aim of “to assess the dietary profiles of vegan, vegetarian, and omnivorous distance runners”.

Answer: We appreciate your concern. However, it is noteworthy that the reason for excluding runners with <50% of CHO intake was the significant association of low-CHO diets with impaired athletic performance (as well as health issues) which could potentially result in compensatory alternations in dietary intake, as supported by references. We should also consider that athletes who consume this or below this limited percentage of CHO (as the dominant fuel for endurance athletes at high intensities during training and racing) usually don’t have performance-related concerns and may have weight-related goals. Therefore, following the evidence-based recommendation available in literature, the team of researchers decided to take into account this limitation for CHO intake and-in order to control for a minimal CHO intake that may adversely correspond to health and exercise performance-applied it as an exclusion criterion in order to increase the reliability and validity of the findings.

6.c. Line 159: remove the word “friendly”

Answer: Thank you for your comment.

Action taken: we removed the term.

  1. Results: As per comment in methods, some of this may need to be revised after post-hoc correction has been applied.

Answer: Thank you for your continuous consideration. We kindly refer the respected reviewer to our previously mentioned answer to your comments 1 and 6; resulting from this, no action is needed.

  1. Discussion:

8.a. Line 289: This comment/sentence is not supported by your results. You did not assess endurance performance, so this cannot be stated. Similarly, the citations used to support this claim are narrative reviews that broadly report on mechanistic benefits of vegan/vegetarian diets. To my knowledge, crossover trials comparing vegan vs. omnivorous diets have shown no differences in performance adaptations across a training program. Please revise.

Answer: We understand your concern. The sentence was an introductory statement for the paragraph (not belonging to the present findings). However, we agree that it can be expressed with more caution.

Action taken: we revised the statement in a more realistic way, and added 3 references (2 new and one already mentioned) to provide a firm support about the concept.

8.b. Line 292: what do you mean “took advantage”? Maybe had a higher proportion of vegetarian and vegan runners?

Answer: Thank you for the question. In the present study we benefitted from having 3 semi-equal study groups of vegans, vegetarians, and omnivores, which cannot be observed in many other studies on vegan and vegetarian athletes. This was due to an additional focus on vegan and vegetarian runners during recruitment of participants in order to conduct the statistical analysis with a highest possible reliability and validity. As you may know, the prevalence of vegan/vegetarian diet is generally about 5-10% in adult populations (a little higher among endurance runners, as mentioned in the Introduction), but in the present study we were able to recruit a considerable number of vegan (n = 76) and vegetarian runners (n = 40) which seems to be comparable to the number of participants in the omnivorous group (n = 95).

Action taken: We revised the statement and added further information to make it clearer.

8.c. Line 308: I think this line is misleading. The difference is 4.5 vs 2.5 vs 2.9 (which is not double both groups). Can you please contextualize this difference in terms of actual alcohol intake (i.e., 1 more alcoholic beverage per week), as I suspect the intake is quite low across all three groups.

Answer: Thank you for your comment. Yes, it was not double, and for this reason we used the term “nearly”. However, we agree that the phrase “nearly twice” should be revised. Regarding the second part of the comment, unfortunately we are not able to present the data in form of serving size, because both amount and frequency of intake were studied simultaneously. Therefore, as is mentioned in the Methods, all data (including those for alcohol intake) are expressed using a heuristic index (as a compound variable) ranging between 0 and 100, in order to scale dietary intake displayed by measures, items, and clusters.

Action taken: We had a small revision in the statement (removing “nearly twice”).

8.d. Line 325: I think you could remove this sentence.

Answer: Thank you. Yes, we agree.

Action taken: The statement was removed.

8.e. Line 330: This addition feels like taking a chance at self-citation to me. I don’t think a single case report strengthens the previous point, which is based off some much larger observational cohorts. Please remove this sentence.

Answer: Thank you for expressing your concern. There was no intention for self-citation as any scientific finding/conclusion can be reported where applicable.

Action taken: The statement was removed and a more-general new statement was added, supported by a new reference.

8.f. Line 354: two points associated with this sentence. Firstly, I don’t think the way it is written makes sense. Secondly, the claim is not supported by the citation. You state that “Recent evidence indicates that distance runners have acceptable health-related behaviours regarding food choices to get the desired ingredients” – and the cite a paper that suggests half-marathon runners are healthier than other runners? This again feels like a sneaky chance to self-cite. Please remove this setnece.

Answer: While we appreciate your concern, it should be mention that the statement was one of the conclusions derived from reference 73, and no any other intention was behind. However, it seems that the statement can be expressed more clearly.

Action taken: The statement was revised, accordingly.

8.g. Line 355: I am not sure the term “healthier” here is appropriate. While refined grains do have associations with disease in various populations, this association mostly disappears when accounting for bodyweight (i.e., health status is dictated by excess adiposity, not be the refined grains themselves). Similarly, replacing saturated fats with various oils has been shown to lower CVD risk – which could be considered “healthier.” Please revise this sentence to better describe your results.

Answer: Thank you for your attentive comment.

Action taken: The statement was revised, accordingly.

8.h. Line 373: please change to “may suggest”

Answer: Thank you.

Action taken: The statement was revised, accordingly.

8.i. Line 412: one limitation that needs to be mentioned is related to sample size. While ~270 runners is nothing to sneeze at, the larger than average number of participants following vegetarian/vegan diets means that this data may not be representative of the larger running population – please state this.

Answer: Thank you for your comment. As is mentioned earlier in our answer to your “general comment”, there is no doubt that a larger sample size increases the representativeness of the results and interpretations, but compared to similar studies the present sample size is rather not small, indeed; however, we did not make a statement of the representativity of our sample or findings. Additionally, we should consider that 106 participants were also excluded from our final analysis in the process of data clearance in order to present more valid findings. Regarding the second concern, we believe that the larger-than-average number of vegetarian/vegan participants can be considered a positive point from statistical viewpoint as there was 3 semi-equal study groups. However, we agree that this is not in line with the normal distribution/prevalence of vegans and vegetarians, and may be mentioned as a limitation.

Action taken: we included this fact among the study limitations.

8.j. Line 416: how/what future intervention studies do you suggest being performed based upon your findings?

Answer: Thank you for your valuable comment. In this regard, it is notable that we have proposed several follow-up studies on different aspects of plant-based diets which are not limited to the NURMI Study data (e.g., current projects: https://www.science2.school/en/ and https://uni.science2.school/en/, as well as future interventional projects that currently we are at their early stages by study design).

Action taken: we completed the statement, accordingly.

  1. Conclusion: Line 427: please revise this sentence and remove the words “healthier” instead state the main results/ areas of difference.

Answer: We are thankful of your comment.

Action taken: We included a summary of the findings and had a revision on the associated conclusion as well.

Reviewer 2 Report

The design and the idea of the investigations is good but there are 2 main issies that must be fixed.

1) Statisitcs

- Comparison between groups was performed using Kruskal-Wallis test. It is a non-parametric test. Did you check normality of distribution of the sample? Otherwise use ANOVA

- How did you allocate the differences between groups if you did not perform post-hoc test?

2) Conclusions are too adventurous.

" The present study may also help to expand personalized dietary strategizing based on diet type and race distance..."

The manuscript only identified differences between diets not which one is better for performance or what what banace between food clusters is more appropriate

3) English expression and spelling must be reviewed by a native speaker of an specialized company.

Example: Line 427. "trailed by vegetarians". Not very scientific

Author Response

Dear Reviewer #2,

Thank you for your consideration, positive feedback, and the opportunity to revise the manuscript nutrients-1835516 entitled “Dietary Intake of Omnivorous, Vegetarian, and Vegan Endurance Runners – Results from the NURMI Study (Step 2)”, which has led to a significant improvement of the manuscript.

In response to your valuable comments, we provided some evidence along with detailed explanations that directly address each comment. We hope our responses below and revisions throughout the manuscript will satisfy the existing concerns of the reviewer and editors. Changes in the manuscript have been highlighted via “Track Changes”.

Kind Regards,

Katharina Wirnitzer on behalf of the team of authors

Review Report 2

-----------------------------------------------------------------------------------------------------------------------------

Comments mentioned by Reviewer #2 (in blue) and our responses to each comment (in red)

-----------------------------------------------------------------------------------------------------------------------------

The design and the idea of the investigations is good but there are 2 main issues that must be fixed.

1.a. Comparison between groups was performed using Kruskal-Wallis test. It is a non-parametric test. Did you check normality of distribution of the sample? Otherwise use ANOVA

Answer: Thank you very much for your attentive comment. However, there seem to appear a basic misunderstanding. With using a scale where the value 0 is impossible, e.g. age and BMI, the Kruskal-Wallis test (non-parametric test and here are no assumptions on the normality of distribution of the sample). We did not perform an ANOVA (non-parametric test and one assumption is the normality of the Residuals and rather not the normality of the sample) but performed a regression analysis in order to test the main research question (difference in dietary intake between subgroups).

1.b. How did you allocate the differences between groups if you did not perform post-hoc test?

Answer: Thank you for your attentive comment. As the NURMI study is an explorative study (generating hypothesis) and not an experimental/interventional study, a post hoc test was not performed (as not required). The further advanced analysis was performed by a regression analysis (forest plot) by testing multiple contrasts with the omnivorous diet as reference (vegetarian and vegan compared to mixed diet). The assessment was made using the parameters of the regression analysis in order to estimate the effects.

1.c. Post hoc is only for intervention studies to test measures pre vs. post intervention!

Answer: Thanks again for your comment. We kindly refer the respected reviewer to our answer to your comment 1b.

  1. Conclusions are too adventurous.

"The present study may also help to expand personalized dietary strategizing based on diet type and race distance..." The manuscript only identified differences between diets not which one is better for performance or what banace between food clusters is more appropriate

Answer: Thank you for your attentive comment. As the previous statements in the section Conclusion have sufficiently expressed the content of this statement (i.e., the relation between “precision sport nutrition” and “the differentiated findings based on diet type“), it seems that this confusing statement can be removed without making any problem in the structure of Conclusion.

Action taken: We removed the statement.

  1. English expression and spelling must be reviewed by a native speaker of an specialized company.

Example: Line 427. "trailed by vegetarians". Not very scientific

Answer: Thank you very much for bringing out this issue to our attention. In fact, the team of authors includes a US-native speaker (a researcher with a scientific background in the area of health and sports science) who proofread the manuscript prior to submission. However, it seems that our crowded schedule during the final weeks of summer semester has slightly affected the quality of submitted manuscript.

Action taken: Following the reviewers advice, we had a new comprehensive proofreading and revised several parts of the manuscript, accordingly.

Round 2

Reviewer 1 Report

General comments:

Thank you for taking the time to address my comments and concerns so thoroughly. I appreciate the detail provided in your responses, and the changes made to the manuscript.

I especially appreciate the slight changes you have made to the introduction and discussion to make the manuscript a little more “balanced.”

Specific Comments:

Line 137: the addition to this sentence is a little unclear. I am not sure what you mean by “accurate” responses. Maybe something like “as they provided answers comparable to runners who compete in half-marathon, marathon, and ultramarathon events.”

Line 292: this sentence reads a little odd. Maybe something like “With the increasing prevalence of athletes who follow vegan/vegetarian diets, it has been suggested that well-designed plant-based diets can be implemented to successfully manage health and endurance performance.”